# Serological Evidence of Influenza D Virus Circulation Among Cattle and Small Ruminants in France

**DOI:** 10.3390/v11060516

**Published:** 2019-06-05

**Authors:** Justine Oliva, Amit Eichenbaum, Jade Belin, Maria Gaudino, Jean Guillotin, Jean-Pierre Alzieu, Philippe Nicollet, Roland Brugidou, Eric Gueneau, Evelyne Michel, Gilles Meyer, Mariette F. Ducatez

**Affiliations:** 1IHAP, Université de Toulouse, ENVT, INRA, UMR 1225, 31076 Toulouse, France; j.oliva@envt.fr (J.O.); amit.eichenbaum25@uga.edu (A.E.); belinjade@gmail.com (J.B.); maria.gaudino@studenti.unipd.it (M.G.); g.meyer@envt.fr (G.M.); 2Laboratoire Départemental Public du Nord, 59651 Villeneuve-D’ascq, France; JEAN.GUILLOTIN@lenord.fr; 3Laboratoire Vétérinaire Départemental de l’Ariège, 0900 Foix, France; jpalzieu@ariege.fr; 4Laboratoire de l’Environnement et de l’Alimentation, 85021 La Roche-sur-Yon, France; philippe.nicollet@vendee.fr; 5Aveyron Labo, 12031 Rodez, France; brugidou@aveyron-labo.fr; 6Laboratoire Départemental de la Côte d’Or, 21017 Dijon, France; eric.gueneau@cotedor.fr; 7Laboratoire Public Conseil, Expertise et Analyse en Bretagne, 35000 Rennes, France; cea.michel@gmail.com

**Keywords:** Influenza D virus, France, cattle, small ruminants, seroprevalence, epidemiology

## Abstract

Influenza D virus (IDV) has first been identified in 2011 in the USA and was shown to mainly circulate in cattle. While IDV is associated with mild respiratory signs, its prevalence is still unknown. In the present study we show that IDV has been circulating throughout France in cattle and small ruminants, with 47.2% and 1.5% seropositivity, respectively. The high prevalence and moderate pathogenicity of IDV in cattle suggest that it may play an initiating role in the bovine respiratory disease complex.

## 1. Introduction

In 2011, a new influenza virus was isolated from pigs with influenza-like symptoms and shared only 50% overall homology to human influenza C virus. This virus was considered as a new genus and named thereafter influenza D virus (IDV) [1]. IDV circulates widely and has been detected in America [2,3,4], Europe [5,6,7,8], Asia [9,10] and Africa [11]. Several studies demonstrated that IDV has a large host range and a higher prevalence in cattle than in swine and other species, suggesting that bovine could be a main host for IDV. The virus or its specific antibodies were also detected in horses [12], small ruminants [13], camels [11] or feral swine [14]. However, the zoonotic potential of IDV is still unclear. The circulation of IDV in Europe is not fully understood but data is available in Luxembourg and Italy with small cohorts tested: 80% and 93% of the tested cattle sera were positive in Luxembourg and Italy, respectively (*n* = 480 and 420 sera tested in each country) [8,15].

Here, we performed a large scale seroprevalence study of IDV in large and small domestic ruminants at a country level. As we aimed to detect IDV antibodies with an individual prevalence limit of 0.1% for cattle and 0.5% for small ruminants with 95% confidence, at least 3000 and 600 sera were needed, respectively.

## 2. Materials and Methods

Five thousand three hundred and seventy-three animal sera (*n* = 33,181,430 and 625 sera coming from *n* = 92, 45, and 13 herds for cattle, sheep and goat respectively, Appendix A) were collected in official veterinary laboratories and at the Veterinary School of Toulouse from five French regions. Most of these sera were initially collected for infectious bovine rhinotracheitis monitoring. The sampling plan was representative of the population taking into account the major cattle-rearing areas including Bretagne, Pays de la Loire, Bourgogne-Franche-Comté, Hauts-de-France and Occitanie. In addition, sera from Occitanie were retrieved from the Veterinary School of Toulouse large animal clinics (*n* = 509). No data was available on history of respiratory diseases in the farms of each region. All the tested animals were older than 1-year-of age and the detection of maternally derived antibodies can therefore be ruled out. The type of sera, localization and years of collection are described in Appendix A. Three controls sera were used: an in-house polyclonal rabbit anti-IDV serum generated by inoculating rabbits with D/Bovine/Nebraska/9-2/2012 subcutaneously (as described in [11]); IDV negative and positive French cattle sera generated during an experimental infection [16]. All sera were treated with receptor destroying enzyme (RDE, Seika) following the manufacturer’s instructions and hemadsorbed on packed horse red blood cells. Hemagglutination Inhibition (HI) assays were performed as previously described [16], with four hemagglutination units of D/bovine/France/5920/2014 and 1% horse red blood cells. Samples with antibody titers ≥1:20 were considered positive. Statistical analyses were carried out using Graph Pad Prism 5.0. A *p* value ≤0.05 was considered significant. A χ2 test was used to compare IDV seroprevalences between species and between French provinces.

## 3. Results

Our serology results demonstrated that IDV circulates throughout the country, in all tested species (Figure 1). We observed a higher seroprevalence in bovine (47.2%, mean geometric titers or GMT: 67) than in small ruminants (1.5%, GMT 27 for ovine and 3.2%, GMT 31 for caprine), all regions combined (*p* < 0.01). In addition, the small ruminants presented low antibody titers (from 1:20 to 1:160) as compared to those observed in cattle (from 1:20 to 1:1280). We observed that sera from all years of collecting (2014–2018 all regions included) were at least seropositive for one serum. We observed differences of serological prevalence between French regions, ranging between 31–70% for bovine, 0–5.5% for ovine and 1.3–5.8% for caprine. These differences were only significant for cattle (*p* < 0.01, χ2 test). For bovine, the highest seroprevalence was observed in Pays de la Loire, and the lowest in Hauts-de-France. The highest seroprevalence for goat and sheep were in Bretagne and Hauts-de-France regions, respectively.

## 4. Discussion

Our results confirm that if French ovine and caprine are susceptible to IDV, as previously shown in the USA, China and Togo [9,13,16], bovines are the main host for IDV, as previously observed in Luxembourg or in the United States [1,5,8]. Whether virological factors (differences in susceptibility of small and large ruminants to IDV) and/or epidemiological factors (breeding systems, decreasing number of mixed breeding farms in France with time, etc.) are responsible for the differences in prevalence is still not known and further studies are needed to understand the mechanism. Further epidemiological and serological studies including a higher number of mixed breeding farms will also be required to understand the potential transmission of IDV between ruminant species.

We also observed differences of seroprevalence between regions only for cattle. This may be partly explained by the breeding systems with high numbers of fattening units of young bulls or veal calves in Pays de la Loire (highest seroprevalence of 70% with GMT of 86) inducing more exchanges and introductions of young animals between farms from several origins. In contrast, regions such as Hauts-de-France (lowest seroprevalence of 31% with GMT of 45) consist mainly of classical breeding dairy or beef farms. In addition, the seroprevalence reported in Luxembourgish cattle (76%–88%) was higher than the seroprevalence observed in France (31%–70%) [8]. Differences in breeding systems and in number of animals per farm may account for the different seroprevalences observed between the two countries.

We previously showed that IDV was detected in France in 2011 [7]. The high seroprevalence we found between 2014 and 2018 either suggests that once IDV is introduced, it seems to spread very efficiently throughout the country, or that the virus may have emerged well before 2011 in France. Archive sera should be screened to figure out when the virus may have really emerged in the country.

Finally, the high seroprevalence of IDV in French cattle, suggesting that most animals seem to have been infected by IDV without the farmers noticing it, and the low frequency of IDV detection in lungs of calves with severe respiratory disease [7] suggest a moderate respiratory pathogenicity of IDV. This is coherent with the recent results of limited pathogenicity of IDV in calves by experimental infections [16,17]. On the other hand, the high seroprevalence of IDV in adults may explain a partial clinical protection of calves in the first weeks of life by the maternally derived antibodies. The role of IDV in bovine respiratory disease is still unclear, but current data indicate that IDV may act as an initiating pathogen as suggested for bovine parainfluenza virus type 3 (BPI-3) and bovine coronavirus (BCoV), both highly prevalent in Europe and inducing only mild clinical signs by experimental infections. IDV, BCoV and BPI-3 were mainly detected in association with other respiratory pathogens during bronchopneumonia in calves. Studies using next generation sequencing showed that IDV was more frequently detected in cattle with respiratory signs than in healthy animals [18,19]. It has also become clear that many respiratory bovine pathogens (including IDV) with high prevalence are more frequently detected together than by themselves [7]. Regular and global viro- and sero-surveillance, combined with epidemiological studies, are thus warranted to better understand the influenza D virus epidemiology and its role in bovine respiratory disease. In addition, co-infection studies should be performed to understand mechanisms behind pathogens interactions (synergies and antagonisms) in relation to the host immune response and disease severity.

## Figures and Tables

**Figure 1 viruses-11-00516-f001:**
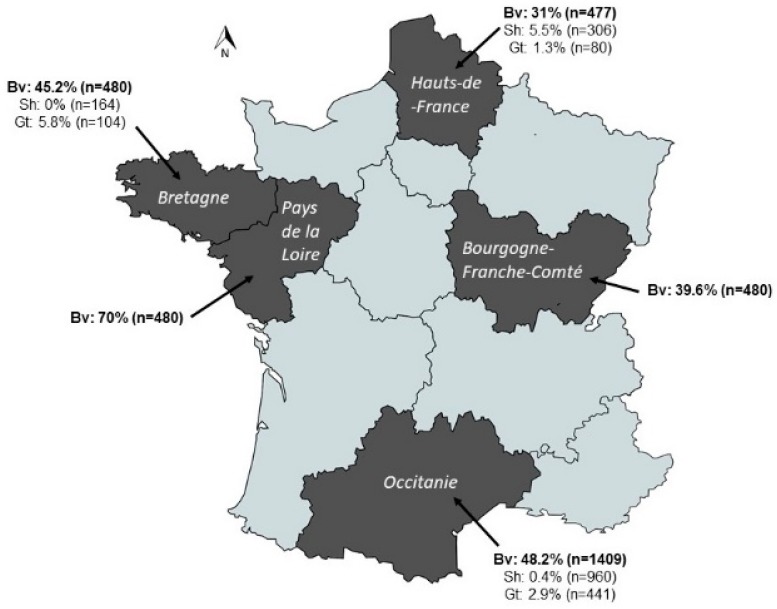
Seroprevalence of influenza D virus in cattle, ovine and caprine from different regions in France. These sera were collected between 2014 and 2018. Selected regions are in black, with their name indicated on the map. Seroprevalence in cattle is indicated in bold font. Bv: bovine; Sh: sheep; Gt: goat.

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
