# Peer review of "Serological Evidence of Influenza D Virus Circulation Among Cattle and Small Ruminants in France"

_viruses, 2019, doi:10.3390/v11060516_

Round 1
Reviewer 1 Report
This paper contains useful additions to the existing literature on the seroprevalence of influenza D virus. I have a comment for your definition, lines 52-53. Some previous papers including a reference 11 defined that antibody titers ≥10 were considered positive. What is the reason that you considered HI titers ≥20 positive in this study? Is there a consistency between this and the previous result?
Figure 1: Compared with the map, the font size for seroprevalence is too small to easily find.
Author Response
We thank Reviewer 1 for the valuable comments and suggestions. We have uploaded our manuscript file using "track changes" to highlight our modifications in the text, as well as a "clean" version (see reply to Reviewer 2).
Reviewer 1 comments and Suggestions for Authors
This paper contains useful additions to the existing literature on the seroprevalence of influenza D virus. I have a comment for your definition, lines 52-53. Some previous papers including a reference 11 defined that antibody titers ≥10 were considered positive. What is the reason that you considered HI titers ≥20 positive in this study? Is there a consistency between this and the previous result?
We agree with the Reviewer that setting a positivity threshold for serology assays for a novel pathogen is somewhat arbitrary. We initially used 10 as a threshold as (i) we had few seropositive animals, (ii) the seropositive animals with titers ≥10 tended to come from the same farms/areas, (iii) there was no real reference to compare our titers to. In the present study, we choose to set the positivity threshold for our HI assay at 20 so that our results can be directly compared with other European studies (references 5,8 and 15 in the manuscript). No difference in data interpretation could be noticed when we changed this threshold to 10 or 40.
Figure 1: Compared with the map, the font size for seroprevalence is too small to easily find.
We have now increased the font size for seroprevalence.
Reviewer 2 Report
In their manuscript Oliva and colleagues investigate seroprevalence of HI antibodies to influenza D virus in cattle and small ruminants in France. The study is interesting and succinct. However, there are several points that need the authors’ attention.
Major points
1) Tables with the data should be uploaded as supplementary figures.
2) It is not clear what was used as positive and negative control? Naïve and known IDV positive sera? Please add this information since controls are really important, specifically for seroprevalence testing.
Minor points
1) Line 24, 25, 26 and throughout the manuscript: It is ‘influenza’, not need to capitalize the ‘i’.
2) Line 42: The same is true for the virus name here.
3) Line 50: ‘inhibition’
4) Table 1: If it is just 1, it should be ‘herd’, not ‘herds’
5) Table 1: What does ‘sera not available’ mean here?
6) Line 63: ‘all years’
7) Line 95: ‘These sera’
8) Line 103: The end of this sentence is weird, please reformulate.
9) Line 110: ‘composed’ sounds wrong here.
10) Line 128: There is a space too much.
11) Line 138: ‘Patents’?????
Author Response
We thank Reviewer 2 for the valuable comments and suggestions. We have uploaded our manuscript file using "track changes" to highlight our modifications in the text, as well as a "clean" version including the supplemental Table and the updated Figure.
Reviewer 2 comments and Suggestions for Authors
In their manuscript Oliva and colleagues investigate seroprevalence of HI antibodies to influenza D virus in cattle and small ruminants in France. The study is interesting and succinct. However, there are several points that need the authors’ attention.
Major points
1) Tables with the data should be uploaded as supplementary figures.
We thank the Reviewer for the suggestion and now have the Table as a supplementary file and not in the main text any longer.
2) It is not clear what was used as positive and negative control? Naïve and known IDV positive sera? Please add this information since controls are really important, specifically for seroprevalence testing.
Indeed, we did not specify the nature of control sera and apologize for the oversight. We used three distinct controls sera: an in-house polyclonal rabbit anti-IDV serum generated by inoculating rabbits with D/Bovine/Nebraska/9-2/2012 subcutaneously (reference 11); IDV negative and positive French cattle sera generated during an experimental infection (reference 16). We have now added this piece of information in the Material and methods section. It reads as follows:
“Three controls sera were used: an in-house polyclonal rabbit anti-IDV serum generated by inoculating rabbits with D/Bovine/Nebraska/9-2/2012 subcutaneously (as described in [11]); IDV negative and positive French cattle sera generated during an experimental infection [16].”
Minor points
1) Line 24, 25, 26 and throughout the manuscript: It is ‘influenza’, not need to capitalize the ‘i’.
We have modified the text as suggested.
2) Line 42: The same is true for the virus name here.
We have modified the text as suggested.
3) Line 50: ‘inhibition’
We have added “HI” in parentheses.
4) Table 1: If it is just 1, it should be ‘herd’, not ‘herds’
The “s” has been removed.
5) Table 1: What does ‘sera not available’ mean here?
We did not receive (and hence did not test) ovine and caprine sera from Bourgogne-Franche-Comté or Pays de la Loire. Actually, there are few small ruminant farms in these 2regions, as compared to the others we looked at.
6) Line 63: ‘all years’
We have added the missing “s”.
7) Line 95: ‘These sera’
We have modified the text as suggested.
8) Line 103: The end of this sentence is weird, please reformulate.
“Whether virological [..] are still to be figured out” was modified to “Whether virological [..] are still not known and further studies are needed to understand the mechanism”.
9) Line 110: ‘composed’ sounds wrong here.
We modified “are mainly composed” by “consist mainly of”.
10) Line 128: There is a space too much.
We modified this.
11) Line 138: ‘Patents’?????
We did not face any intellectual property issue for the present study (no patent indeed). The sera belong to the regional veterinary laboratories.
